# Flow Patterns and Particle Residence Times in the Oral Cavity during Inhaled Drug Delivery

**DOI:** 10.3390/ph15101259

**Published:** 2022-10-13

**Authors:** Brenda Vara Almirall, Kiao Inthavong, Kimberley Bradshaw, Narinder Singh, Aaron Johnson, Pippa Storey, Hana Salati

**Affiliations:** 1Mechanical & Automotive Engineering, School of Engineering, Royal Melbourne Institute of Technology University, Bundoora, VIC 3083, Australia; 2Department of Otolaryngology, Head and Neck Surgery, Westmead Hospital, Westmead, NSW 2145, Australia; 3Sydney Medical School, Faculty of Medicine & Health, The University of Sydney, Sydney, NSW 2006, Australia; 4Department of Otolaryngology-Head and Neck Surgery & Department of Rehabilitation Medicine, Grossman School of Medicine, New York University, New York, NY 10017, USA; 5Department of Radiology, Grossman School of Medicine, New York University, New York, NY 10016, USA

**Keywords:** CFD modeling, oral cavity, respiration, targeted drug delivery, SBES

## Abstract

Pulmonary drug delivery aims to deliver particles deep into the lungs, bypassing the mouth–throat airway geometry. However, micron particles under high flow rates are susceptible to inertial impaction on anatomical sites that serve as a defense system to filter and prevent foreign particles from entering the lungs. The aim of this study was to understand particle aerodynamics and its possible deposition in the mouth–throat airway that inhibits pulmonary drug delivery. In this study, we present an analysis of the aerodynamics of inhaled particles inside a patient-specific mouth–throat model generated from MRI scans. Computational Fluid Dynamics with a Discrete Phase Model for tracking particles was used to characterize the airflow patterns for a constant inhalation flow rate of 30 L/min. Monodisperse particles with diameters of 7 μm to 26 μm were introduced to the domain within a 3 cm-diameter sphere in front of the oral cavity. The main outcomes of this study showed that the time taken for particle deposition to occur was 0.5 s; a narrow stream of particles (medially and superiorly) were transported by the flow field; larger particles > 20 μm deposited onto the oropharnyx, while smaller particles < 12 μm were more disperse throughout the oral cavity and navigated the curved geometry and laryngeal jet to escape through the tracheal outlet. It was concluded that at a flow rate of 30 L/min the particle diameters depositing on the larynx and trachea in this specific patient model are likely to be in the range of 7 μm to 16 μm. Particles larger than 16 μm primarily deposited on the oropharynx.

## 1. Introduction

The human upper airway serves as a carriageway allowing air to pass into the lungs. During inhalation, airborne particulates are transported through the airway anatomy, where high inertial particles impact the posterior oropharynx wall. From here, the $90^{\circ }$ bend at the oral-to-pharynx transition serves as an anatomical defense system to prevent foreign particles from penetrating deeper towards the lungs. Thus, while inhaled airborne particulates from the environment pose a health risk, the pharmaceutical industry seeks to overcome the airway defense system to provide effective pulmonary drug delivery, a critical therapy method for treating respiratory diseases. Furthermore, resistance through the oral airways is much lower than the nasal airways, and this presents an opportunity for pulmonary drug delivery via inhalation through the oral route.

In vitro and in vivo experimental studies of oral airway drug delivery to determine inhaled aerosol deposition have been an ongoing focus for many years [1,2,3,4,5,6,7]. Studies with humans are invasive and challenging; therefore, simulations using mouth–throat models of realistic airways produced from scan data are increasingly used.

The authors in [8], among others [9,10,11], quantified micron particle deposition in realistic oropharyngeal airway replicas of children and later in adults [12], and summarized the deposition data for child airways with a best fit equation that used the inertial parameter term $d_{a}^{2}Q$ ($d_{a}$ is the particle aerodynamic diameter, and Q is the flow rate), suggesting particle deposition occurs due to inertial impaction. Similarly, Ref. [13] correlated deposition efficiency in the oral cavity for 0.93 to 30 μm particles to the particle Stokes number, which also represents impaction as the dominant deposition mechanism. Thus, the primary particle deposition mechanism is well established as inertial impaction [14].

To understand the causes of early deposition from impaction, studies investigated the effects of different devices and mouthpieces. Lin et al. [15] examined the effect of mouthpiece diameter for deposition of 2, 4, 8 μm particles in adult human oral-pharyngeal laryngeal airway cast models at inspiratory flow rates of 30, 60, 90, and 120 L/min, with the suggestion of an optimal combination of 4 μm particles with inspiratory flow rates of ≥60 L/min for adequate drug delivery to the lungs. The authors in [7] determined the deposition from six different pharmaceutical aerosol inhalation devices in an idealized mouth and throat geometry and concluded that the inhaler geometry that the aerosol passes through prior to entering the mouth and throat region can greatly affect the deposition in the mouth-throat. Similarly, Ref. [3] found chlorofluorocarbon (CFC) propellant produced high inertial deposition of 78% in the oropharyngeal region of a human airway replica model compared with hydrofluoroalkane (HFA) propellant with 56% deposition. In [16], CFD was combined with experiments to compare the deposition of aerosols in a realistic model of oral airways, and it was found that DPI devices produced two times more deposition than MDIs in the oral and pharyngeal region. The authors in [17] investigated the effects of mouth–throat geometrical factors, which included the oral cavity volume, glottis area, airway curvature, and total airway volume on particle deposition and showed that realistic airway models significantly influenced mouth–throat deposition compared with the idealized induction port model, with up to a 55% difference. The glottis area and total airway volume were found to be the two most predominant factors.

Computational studies of oral drug delivery employing CFD can provide complementary information through more complex cases for investigations, adding further insight into deposition mechanisms [18,19,20,21,22,23,24,25,26,27,28]. For example, Ref. [29] used the k-ω-SST model and investigated nanoparticle deposition in an extrathoracic oral airway; Ref. [30] applied helical fluid–particle flow dynamics for controlling micron particle deposition in a representative human upper lung–airway model; Ref. [31] investigated the magnetophoretic steering of microsphere carriers of nanoparticles through the oral airway; Ref. [32] compared the effects of using helium–oxygen and air for particle deposition in a realistic model of human oral extrathoracic; Ref. [33] investigated enhanced condensational growth applied to respiratory drug delivery with comparisons to in vitro data, similar to [34], which investigated dry powder inhaler aerosol deposition in a model of tracheobronchial airways to cover the limitations of their prior experimental investigations.

In this study, we apply CFD to further investigate the aerodynamics of inhaled particles by characterizing the inhaled air as the carrier phase transporting the particles. We determine the particle residence times through the airway, which may help to optimize the administration delivery techniques, e.g., spacers and breathing profiles, and quantify the local deposition fractions along different regions of the airway anatomy.

## 2. Results

### 2.1. Flow Pattern

Figure 1 demonstrates the velocity contour and vectors and turbulent kinetic energy contours at different coronal planes and the mid-sagittal plane for the breathing rate of 30 L/min. The air passing through the mouth shows a concentrated region in the middle of the oral cavity where air is drawn into a core region medially, and superiorly shown in the velocity contour, and is supported by the position of the largest vectors in Figure 1a. The lateral sides of the oral cavity produce secondary flow where Dean vortices form.

At the posterior oral cavity, the airway shape becomes more circular, and the airflow experiences a strong mixing effect that will enhance deposition of inertial particles. Continuing with the flow direction to the oropharynx, the geometry follows a 90${}^{\circ }$ bend, while at the same time the cross-sectional area reduces to a minimum cross-section at *p5*. As a result, the flow accelerates with a sharp increase in velocity to its peak values. Large inhaled particles are expected to be captured at this bend between *p4* and *p5* through inertial impaction. Flow through the oropharynx can therefore be characterized similarly to a 90${}^{\circ }$ bent pipe.

At cross-section *p6* (the pharynx), the airway cross-section expands, but the accelerated flow from *p5* continues to move through as a high-velocity jet along the posterior wall. Peak velocity is found at more than 10 m/s, as well as peak turbulent kinetic energy (TKE) at $50\;m^{2}/s^{2}$ (Figure 1c). The local accelerated flow from *p6* surges through the larynx forming the so-called laryngeal jet.

### 2.2. Vortex Structures

The pharyngeal jet leading into the larynx disrupts the constant stream of airflow to a highly disturbed jet flow with unsteady characteristics, as seen in Figure 2. The flow structures were visualized using the isosurface of the *q*-criterion $=3\times 10^{6}$ s${}^{2}$. This highly turbulent region, characterized by vortex structures, will further enhance the deposition of larger micron particles, although the smaller micron particles may pass through with the jet flow momentum. A maximum velocity was reached at the site of the minimum cross-sectional area in the oropharynx. The flow decelerated further downstream due to the expansion of the geometry, and a flow recirculation appeared downstream of the separation point. To emphasize this flow phenomenon, the *q*-criterion is colored using the *z*-velocity, where the positive values show the upward motion of the recirculation in Figure 2a.

Vortices were first produced at the oropharynx (oropharyngeal jet) from the change in geometry, namely, the bend and the significant decrease in cross-sectional area. The vortices were generated due to shear layer instabilities interacting with the laryngeal jet. The vortices are colored by local velocity magnitude, which correlates high velocity with an increase in vortex structures, specifically in the larynx region, with a higher velocity at the posterior wall of the larynx (laryngeal jet). The laryngeal jet started to weaken, transforming into uniform flow in the trachea, where the cross-sectional area remains almost constant.

### 2.3. Particle Trajectories

A Appendix A is provided to support the visualisation found as ‘*Appendix A*’ in the Appendix A. Figure 3 shows the particle trajectories colored by particle diameter for the reported residence time between the initial particle release at t=0 s to t=0.22 s. In the oral cavity, the larger particles (colored red in Figure 3) are located in the middle of the particle stream that pushed superiorly as the inhaled air carries them to the back of the throat.

The smaller particles (colored blue) are more disperse and can be seen around the larger particles. The sudden expansion in the cross-sectional area from the lips to the oral cavity caused flow separation from the bottom lip. The separated flow causes the air to move superiorly, which transports the particles to the same location.

At t=0.07 s, the particles first reach the back of the throat, and at this point we witness the effect of inertial impaction of the largest (red-colored) particles, approximately >22 μm in diameter, depositing onto the oropharnyngeal, where the geometry exhibits a 90${}^{\circ }$ bend. The finer particles (colored blue) of approximately <12 μm reach the larynx after 0.12 s.

Figure 4 shows the particle trajectories from t=0.27s to t=0.46 s, where the particle stream becomes thinner as fewer particles are moving through. These “late” particles were initially located furthest from the mouth inlet. The top view shows a very narrow stream as particles are drawn down the center of the oral cavity. The particles all complete their trajectories within 0.5 s, where the last image at t=0.46 s shows a small number of particles scattered around the trachea and geometry outlet.

Figure 5 shows the particle penetration of the 50 particles from their initial location. Highlighted in this figure is the difference in time (x-axis) for particles to reach their terminal velocity. Some of the particles furthest away at a distance of 0 to 0.02 mm had not moved in the first 0.07 s of the 0.5 s inhalation, while the particles closest to the inlet quickly reached their terminal velocity within the first 0.07 s.

### 2.4. Particle Deposition Fractions

Figure 6 shows the deposition fraction (DF) of each particle diameter for different anatomical regions at different time intervals. The DF represents the fraction of the deposited particles over the total injected particles. Larger particles (23 μm to 26 μm) mainly deposited in the oral cavity. The maximum deposition fraction of 26-micron particles occurred at 0.1 s. In contrast, the DF of finer particles (7 μm and 10 μm) was low in the oral cavity and was maximum in the larynx at 0.15 s. This is due to flow circulation in the laryngeal region, which impacts the motion of smaller particles. With increasing particle size, thus particle inertia, the filtering effect of the oropharynx bend increases and more particles are deposited.

The DF did not vary significantly for larger particles versus time in the trachea. Particle deposition in the range of 7 to 20 μm increased in the trachea within 0.15 s. The trachea has the lowest deposition due to the deposited particles in the upstream regions. The rest of the particles reached the trachea region’s interior, where the flow was uniform (compared with the larynx) and followed the airflow towards the lungs. Overall, peak deposition of particles occurred between 0.1 s to 0.15 s in the mouth–throat region.

## 3. Discussion

Drug delivery of inhaled particles through the mouth–throat region must overcome the highly complex airway geometry. For pulmonary drug delivery, particles must navigate the tortuous geometry without impacting the airway walls that are lined with wet mucus, while the larynx and trachea are targeted in cases of laryngeal–tracheal stenosis or inflammation in the larynx. This study demonstrated the airflow patterns under the influence of a patient-specific mouth–throat airway geometry. A narrow central air core stream was generated in the oral cavity, where particles were drawn in. This may explain a high propensity for particle deposition in the roof of the oral cavity. The oropharyngeal bend was implicated for its role in filtering out larger micron diameter particles (approximately, >20 μm), and this explains the associated sense of taste of the drug formulation when spray formulations are delivered.

The particle residence times showed that the drug delivery event was completed within 0.5 s for particles released within a 3 cm-diameter sphere in front of the oral cavity. The inhaled air velocity influence was constrained within the so-called breathing region, identified as an approximately 2–3 cm region from the airway openings [35,36], which was evident in this study. The particles furthest away from the oral cavity took notably longer to reach their terminal velocity with the inhaled air. Nevertheless, all particles were inhaled and either deposited or escaped into the deeper trachea and lung regions.

The short inhaled particle event of 0.5 s makes coordinating breathing profiles (sniff, shallow, and exhale) with drug delivery actuation very challenging, in particular when spacers are used as additional attachments to metered-dose inhalers (MDI) or dry powder inhalers (DPI).

The deposition fraction analysis provides insight into optimal particle diameters for specific anatomical regions. While a single flow rate was used in this study, one can rely on the particle Stokes behavior to extrapolate the combination of flow rate and particle diameter through the inertial parameters of $d_{a}^{2}Q$, where $d_{a}$ is the aerodynamic diameter and *Q* is the flow rate.

## 4. Materials and Methods

### 4.1. Model Construction

An MRI scan was performed on a male (47 years old, 79 kg) with a Siemens 3T Magnetom Prisma Fit scanner with the following parameters: scanning sequence = Gradient Recall, sequence variation: Spoiled, slice thickness = 0.5 mm, repetition time = 6.7 ms, and echo time = 3.05 ms. Sample images of the MRI scan are shown in Figure 7a. The DICOM files were imported into a 3D slicer, and airway segmentation of individual anatomical regions of interest was performed by a trained ear, nose, and throat clinician. The segmentation produced a 3D volume computer model, and this was manually refined by reducing the effects of noise and smoothing unrealistic regions. External facial features (e.g., lips) were included to ensure realistic inhalation entering the oral cavity [35]. To prevent large flow gradients forming at the nasopharynx outlet boundary, an artificial extension (50 mm in length) was added. The final model is shown in Figure 7 with labeled anatomical regions and ten cross-section planes for flow visualization.

### 4.2. Fluid Phase Modeling

A CFD simulation was performed for a constant inhalation flow rate of 30 L/min. In previous particle deposition studies [21,24], three flow rates are typically evaluated, 15, 30, and 60 L/min; we selected the 30 L/min as the median value for the flow field to represent the inhalation flow rate. The influence of airflow on particle impaction is described by the inertial parameter that incorporates the different flow rates, and therefore variations in the flow rate can be deduced using the inertial parameter.

At 30 L/min, we assume the presence of turbulence, particularly due to the larnygeal jet, and the Stress-Blended Eddy Simulation (SBES) turbulence model was used to account for the turbulent flow. The SBES model is a hybrid RANS–LES model that overcomes the highly restrictive LES limitation of very fine meshes in the wall boundary layer by integrating to the wall using a RANS model. The k-ω SST (Shear Stress Transport) model was used with the LES Wall-Adapting Local Eddy-Viscosity (WALE) model for the scale resolving component. The incompressible flow equations describing the conservation for mass and momentum are expressed as:(1)$$\begin{array}{cc} \frac{\partial }{\partial x_{i}}\left({\bar{u}}_{i}\right) & =0 \end{array}$$(2)$$\begin{array}{cc} \frac{\partial }{\partial t}\left(\rho {\bar{u}}_{i}\right)+\frac{\partial }{\partial x_{j}}(\rho {\bar{u}}_{i}{\bar{u}}_{j}) & =-\frac{\partial \bar{p}}{\partial x_{i}}+\frac{\partial \sigma_{ij}}{\partial x_{j}}+\frac{\partial \tau_{ij}}{\partial x_{j}} \end{array}$$
where $u_{i}$ is the flow velocity vector, ρ is the fluid density, p is the pressure, and $\sigma_{ij}$ is the stress tensor due to molecular viscosity. The overbar $\bar{\varphi }$ on the scalar quantity φ denotes a Reynolds-averaging operation in the RANS formulation and a spatio-temporal filtering operation in the LES formulation. The turbulence stress tensor $\tau_{ij}\equiv \rho \left(\bar{u_{i}u_{j}}-{\bar{u}}_{i}{\bar{u}}_{j}\right)$ for the SBES formulation is defined to blend between the Reynolds stress tensor $\tau_{ij}^{\mathrm{RANS}}$ for the RANS formulation and the subgrid-scale stress tensor $\tau_{ij}^{\mathrm{LES}}$ for the LES formulation, according to the blending function:(3)$$\tau_{ij}=f_{s}\tau_{ij}^{\mathrm{RANS}}+\left(1-f_{s}\right)\tau_{ij}^{\mathrm{LES}}$$
where $0\leq f_{s}\leq 1$ is the shielding function.

### 4.3. Meshing and Time-Step Requirements

The LES component of the SBES model required a sufficiently fine mesh to resolve the large eddies. The model was meshed with a polyhedral mesh on its surface, an then filled with hexahedrons in the bulk flow region, and this mesh was connected to the inflation with polyhedra cells. Six prism layers were placed on the walls where the normalized wall distance was $y^{+}<1$ on all walls. The strict requirements of normalized wall distances $x^{+}$ and $z^{+}$ were not required for the SBES model. The final meshed model contained 4.7 million cells, with the internal hexahedral cells having lengths of Δ=0.3 mm. To ensure sufficient spatial resolution was achieved, the mesh was evaluated based on the turbulence integral length scale, which includes most of the energy-containing eddies and is estimated from a precursor k-ω SST model simulation using:(4)$$l_{o}=\frac{k^{1.5}}{\varepsilon }=\frac{k^{0.5}}{C_{\mu }\omega },\;\mathrm{where}\;C_{\mu }=0.09$$
where we achieved 70–80% of the turbulence kinetic energy by resolving the largest eddies. The time-step size also needs to resolve the energy-containing eddies. An estimate of the time resolution of the model was calculated based on the time scale, defined as:$$\lambda_{g}/u^{\prime }={\left(15\nu /\varepsilon \right)}^{1/2}=\sqrt{15}\tau_{\eta }$$
where a time step of $\Delta t=1.0\times 10^{-4}$ s resolved over 70% of the flow field, which was every region except the larynx. A smaller value of $\tau_{\eta }=2.5\times 10^{-5}$ s was used to ensure the time resolution was sufficient to establish a scale-resolved flow field. A precursor steady-state kω SST simulation was performed to establish the flowfield. The simulation was then switched to transient flow with the SBES model and simulated for 0.03 s before the particle phase was introduced.

### 4.4. Particle Modeling

Particles were modeled using the Discrete Phase Model, which simultaneously tracks individual particles through the flow field using an equation of motion given by:(5)$$\begin{array}{c} m_{p}\frac{d\vec{u_{p}}}{dt}=m_{p}\frac{\vec{u}-\vec{u_{p}}}{\tau_{r}}+m_{p}\frac{\vec{g}\left(\rho_{p}-\rho_{g}\right)}{\rho_{p}}+\vec{F} \end{array}$$
where $m_{p}$ is the particle mass, $\vec{u}$ is the fluid phase velocity, $\vec{u_{p}}$ is the particle velocity, $\rho_{g}$ is the gas (air) density, $\rho_{p}$ is the density of the particle, $\vec{F}$ is an additional force, $m_{p}\frac{\vec{u}-\vec{u_{p}}}{\tau_{r}}$ is the drag force, and $\tau_{r}$ is the droplet or particle relaxation time calculated by:(6)$$\begin{array}{c} \tau_{r}=\frac{\rho_{p}d_{p}^{2}}{18\mu }\frac{24}{C_{d}Re} \end{array}$$
here, μ is the molecular viscosity of the fluid, $d_{p}$ is the particle diameter, and Re is the relative Reynolds number, which is defined as:(7)$$\begin{array}{c} Re=\frac{\rho d_{p}\left|\vec{u_{p}}-\vec{u}\right|}{\mu } \end{array}$$

Moreover, the drag coefficient is taken from [37] and defined by:(8)$$\begin{array}{c} C_{d}=a_{1}+\frac{a_{2}}{Re_{p}}+\frac{a_{3}}{Re_{p}} \end{array}$$
where $a_{1}$, $a_{2}$, and $a_{3}$ are empirical constants for smooth spherical particle over different ranges of particle Reynolds number. The particle Stokes number is:(9)$$\begin{array}{c} St=\tau \frac{U_{o}}{L_{o}} \end{array}$$
$U_{o}/L_{o}$ is the fluid time scale that is defined by the ratio of a characteristic velocity to a characteristic length.

Seven monodisperse particles with aerodynamic diameters ($\rho_{p}$ = 1000 kg/m${}^{3}$) of 7, 10, 13, 16, 20, 23, and 26 microns were introduced in front of the open mouth in a spherical shape with a radius of 0.015 m (Figure 8). Ten particles of each diameter were introduced from t=0.03 s to t=0.08 s at increments of the flow step of $\Delta t=2.5\times 10^{-5}$ s, resulting in a total of 140,000 particles.

## 5. Conclusions

Inhaled drug delivery via the mouth–throat airway has been well studied for targeting the lung region. While it is well accepted that micron particles under high flow rates are susceptible to inertial impaction on anatomical sites, knowledge of particle residence times, flow patterns, and the particle deposition event can provide insight into ways to overcome the unintended inertial impaction deposition. This study revealed the time taken for particle deposition to occur was 0.5 s for inhaled particles within a 3 cm-diameter sphere in front of the oral cavity inlet. A narrow stream of particles directed superiorly during inhalation was produced due to flow entrainment induced by the oral cavity geometry. Larger particles >20 μm were found in the central core of the particle stream, which all impacted the oropharynx. The smaller particles <12 μm were more dispersed throughout the oral cavity and could navigate the curved geometry and laryngeal jet, escaping through the tracheal outlet. Limitations of this study include a smooth rigid airway wall geometry, where in fact during inhalation the airway moves, although the short particle residence time of 0.5 s may render the airway wall motion less significant. The airway walls also lacked a wet mucus layer, which may influence deposition rates.

## Figures and Tables

**Figure 1 pharmaceuticals-15-01259-f001:**
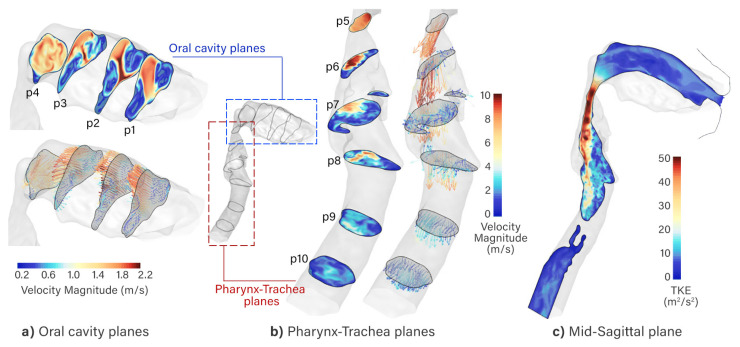
(**a**) Velocity contours at coronal planes where the velocity color range is 0 to 2.2 m/s. (**b**) Velocity contours of sagittal plane where the velocity color range is 0 to 10 m/s. (**c**) Turbulent kinetic energy contour in the mid-sagittal plane.

**Figure 2 pharmaceuticals-15-01259-f002:**
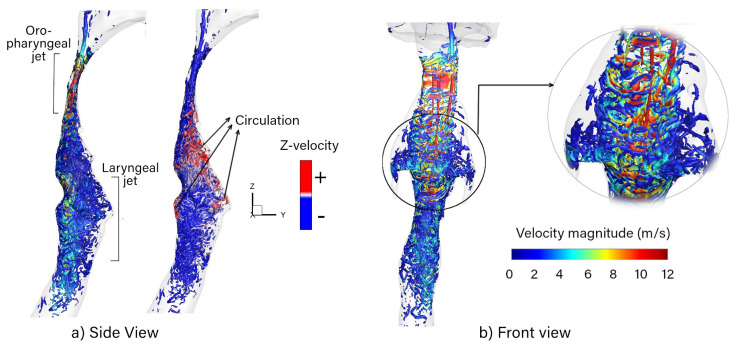
Vortex structures in the mouth–throat airway visualized by *q*-criterion at values of $=3\times 10^{6}$ s${}^{2}$. (**a**) Side view of the vortices colored by velocity magnitude and *z*-velocity (vertical axis). (**b**) Front view depicting vortex structures filling the entire oropharyngeal and laryngeal regions.

**Figure 3 pharmaceuticals-15-01259-f003:**
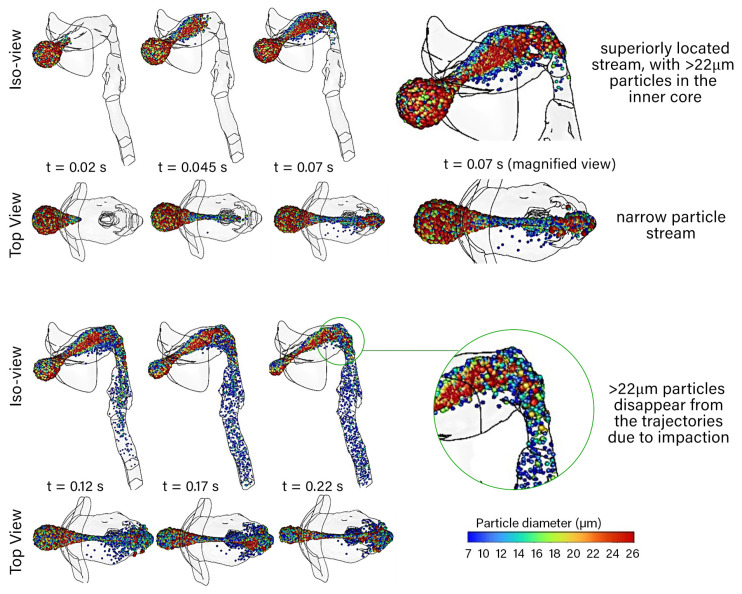
Isometric and top view of the mouth–throat showing the particle trajectories at three time increments from the start of the initial particle release at t=0 s to t=0.22 s. The particles are colored by diameter in microns.

**Figure 4 pharmaceuticals-15-01259-f004:**
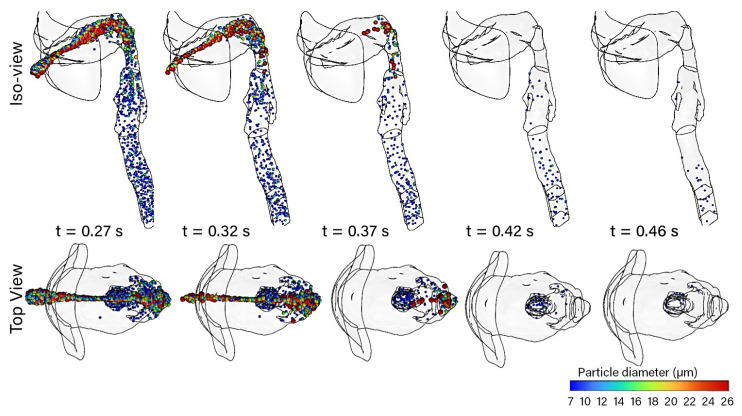
Isometric and top view of the mouth–throat, showing the particle trajectories at five time increments from t=0.27 s to t=0.46 s. The particles are colored by diameter in microns.

**Figure 5 pharmaceuticals-15-01259-f005:**
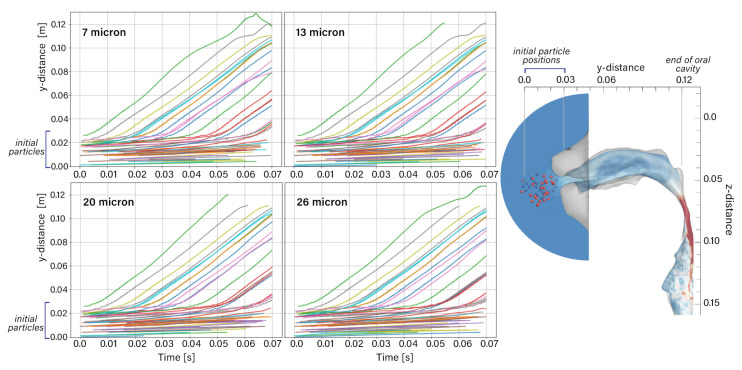
Particle penetration in the streamwise direction over the time period of t=0.0 s to t=0.07 s.

**Figure 6 pharmaceuticals-15-01259-f006:**
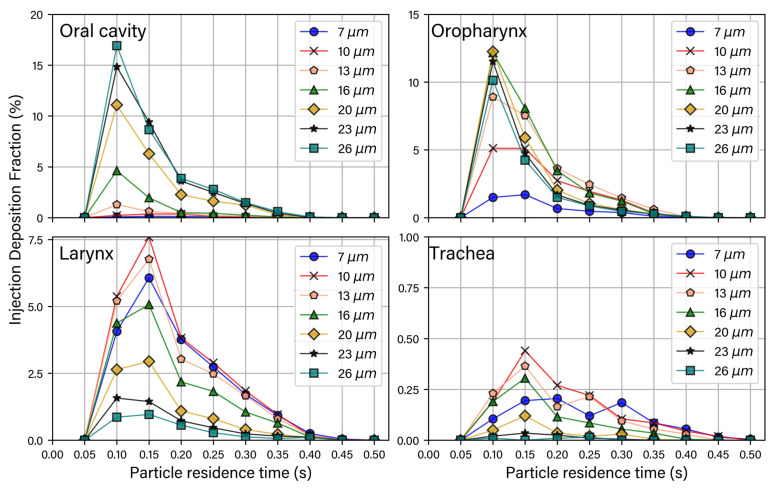
Deposition fraction for each injection at different time intervals (time intervals = 0.5 s).

**Figure 7 pharmaceuticals-15-01259-f007:**
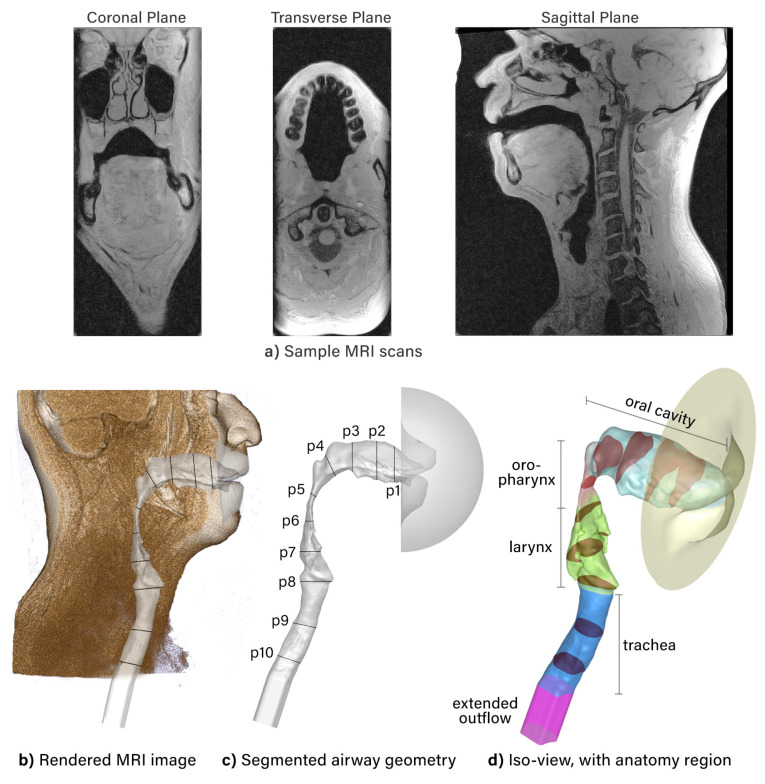
Airway geometry. (**a**) Sample MRI scans depicted in the coronal, transverse, and sagittal planes. (**b**) Three-dimensional rendered image from MRI scans with the segmented airway overlayed. (**c**) Segmented airway geometry labeled with cross-sectional planes (p1 to p10) from the oral cavity to the trachea. (**d**) Isometric view of the airway model showing the shape of the cross-section planes. Separated anatomical regions are colored as follows: light blue = oral cavity; red = oropharynx; green = larynx; dark blue = trachea; pink = extended outflow.

**Figure 8 pharmaceuticals-15-01259-f008:**
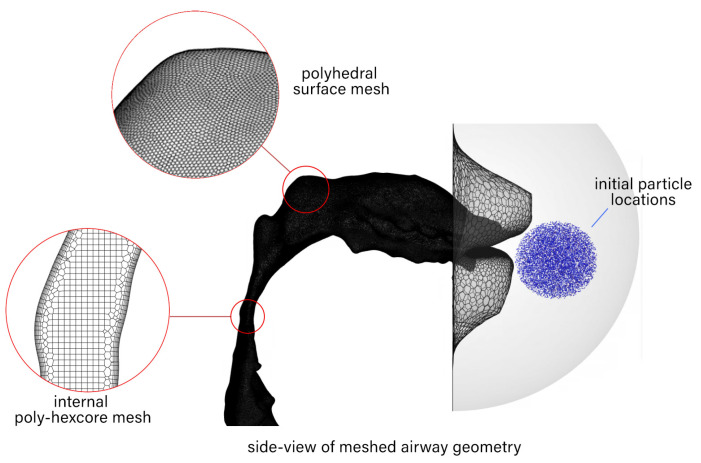
Side view of airway geometry depicting the surface, internal mesh, and the initial particle locations.

## Data Availability

Data are contained within the article and Appendix A.

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
