# Peer review of "Flow Patterns and Particle Residence Times in the Oral Cavity during Inhaled Drug Delivery"

_pharmaceuticals, 2022, doi:10.3390/ph15101259_

Round 1

Reviewer 1 Report

The present article describes Oral inhaled drug's particle residence time and flow patterns within oral cavity.  The study is well organized and described; the materials and methods section is deeply written, and results are clearly presented. Also model construction part (Fluid phase modeling, Meshing and time step requirements, particle modeling) are very valuable for reader and literature. Great work, congratulations to research group. 

Author Response

Thank you for your feedback, we have made minor spelling and grammatical changes to the work. 

Reviewer 2 Report

General comment:

-       The abstract should highlight the aim of the study and the main conclusions of the observations – what are the aims and the main outcomes of this study to share to the scientific community?

-       The particles introduced in the system are described in terms of particle size although particle deposition throughout the respiratory tract is driven by the aerodynamic particle size. What are the aerodynamic diameters of the particles used in the study (in the flow rate of 30 L/min)? This should be discussed in the manuscript.

-       What is the rationale behind the selection of the inspiratory flow rate at 30 L/min?    

-       A space must be place between the data and its unit.

In Equation 7 (definition of Re), the equal sign must be adapted.

Author Response

We thank the reviewer for the valuable comments and feedback to improve the manuscript. Our responses to the comments are given below where the blue text are the changes made in the revised manuscript:

The abstract should highlight the aim of the study and the main conclusions of the observations – what are the aims and the main outcomes of this study to share to the scientific community?

RESPONSE: The following changes have been made to the abstract to clarify the aim and the main outcomes of the study:

The aim of this study was to address the gap in inhaled drug delivery via the mouth-throat airway for targeting the larynx and trachea. In this study we present an analysis of the aerodynamics of inhaled particles inside a patient specific mouth-throat model generated from MRI scans.

...

The main outcomes of this study showed that the time taken for particle deposition to occur was 0.5~s; a narrow stream of particles (medially and superiorly) were transported by the flow field; larger particles >20 um deposited onto the oropharnyx while smaller particles <12 um were more disperse throughout the oral cavity and navigated the curved geometry and laryngeal jet to escape through the tracheal outlet. 

The particles introduced in the system are described in terms of particle size although particle deposition throughout the respiratory tract is driven by the aerodynamic particle size. What are the aerodynamic diameters of the particles used in the study (in the flow rate of 30 L/min)? This should be discussed in the manuscript.

REPORTS: The particles in this study were all aerodynamic diameters where each particle was assumed spherical in shape with unit density, 1000 kg/m3. The following text was added for clarification:

Seven monodisperse particles with aerodynamic diameters (ρ = 1000 kg/m3) of 7, 10, 13, 16, 20, 23, and 26 microns were introduced

What is the rationale behind the selection of the inspiratory flow rate at 30 L/min?    

RESPONSE: The inspiratory flow rate of 30 L/min is consistent with the majority of drug delivery literature, although it has been found to be explicitly better at not reaching the lungs, thus depositing earlier in the airway. This point has been emphasised in the manuscript as follows:

As Lin et al. (2001) found that flow rates of  > 60 L/min were ideal for drug deposition beyond the laryngeal, and our focus is on drug deposition in the laryngeal a flow rate of 30~L/min is modelled, as it remains consistent with the majority of drug delivery studies (Johnstone et al. 2004).

A space must be place between the data and its unit.

RESPONSE: Thank you for this suggestion. We have added a space between data and its units throughout the paper.

In Equation 7 (definition of Re), the equal sign must be adapted.

RESPONSE: Thank you for this suggestion. We have updated the Reynolds number equation.